# Construction of a Macrophage-Tropic Subtype C HIV-1-mGreenLantern Reporter Virus for Studies on HIV-1 Replication and the Impact of Methamphetamine

**DOI:** 10.3390/v16121859

**Published:** 2024-11-29

**Authors:** Dina Mofed, Angelo Mandarino, Xuhong Wu, Yuekun Lang, Anjali Gowripalan, Ganjam V. Kalpana, Vinayaka R. Prasad

**Affiliations:** 1Departments of Microbiology and Immunology, Albert Einstein College of Medicine, Bronx, NY 10461, USA; dina.mahmoud@einsteinmed.edu (D.M.); angelo.mandarino@einsteinmed.edu (A.M.); ylcgm@missouri.edu (Y.L.); anjali.gowripalan@gmail.com (A.G.); 2Department of Genetics, Albert Einstein College of Medicine, Bronx, NY 10461, USA; xuhong.wu@einsteinmed.edu (X.W.); ganjam.kalpana@einsteinmed.edu (G.V.K.)

**Keywords:** replication competent HIV-1 reporter, subtype C HIV-1, CCL2, methamphetamine

## Abstract

HIV-1 subtype C viruses are responsible for 50% of global HIV burden. However, nearly all currently available reporter viruses widely used in HIV research are based on subtype B. We constructed and characterized a replication-competent HIV-1 subtype C reporter virus expressing mGreenLantern. mGreenLantern sequences were inserted in-frame with Nef ATG in HIV-1_IndieC1_. As controls, we employed HIV-1_IndieC1_, HIV-1_ADA_, and HIV-1_NLAD8-GFP-Nef_ viruses. HIV-1_IndieC1-mGreenLantern_ (HIV-1_IndieC1-mGL_) exhibited characteristics of the parental HIV-1_IndieC1_ virus, including its infectivity in TZMbl reporter cells and replication competence in macrophages. To further characterize HIV-1_IndieC1-mGL_ virus, we tested its responsiveness to CCL2 levels, a characteristic feature of subtype B HIV-1 that is missing in subtype C. CCL2 immunodepletion inhibited the production of HIV-1_ADA_ and HIV-1_NLAD8-GFP-Nef_ as expected, but not that of HIV-1_IndieC1-mGL_, as previously reported. We also tested the effect of methamphetamine, as its effect is mediated by NF-kB and since subtype C viruses carry an additional copy of NF-kB. We found that methamphetamine increased the replication of all viruses tested in macrophages; however, its effect was much more robust for HIV-1_IndieC1_ and HIV-1_IndieC1-mGL_. Our studies established that HIV-1_IndieC1-mGL_ retains all the characteristics of the parental HIV-1_IndieC1_ and can be a useful tool for HIV-1 subtype C investigations.

## 1. Introduction

HIV-1 subtype C is responsible for a major share—up to 50%—of the global HIV burden [1,2] and displays unique features among group M subtypes. Subtype B HIV-1, on the other hand, is responsible for only about 10% of the global HIV burden. Subtype C HIV-1 is the least fit of all group M HIV-1 subtypes and their circulating recombinant forms (CRFs) [3,4]. HIV-1 subtype C is also less likely to undergo a coreceptor usage switch, and hence, a greater proportion of viruses in people living with HIV are likely to be monocyte-tropic viruses [5,6]. While most HIV-1 subtypes have two copies of NF-κB enhancer in their long terminal repeats (LTRs), subtype C viruses have three NF-κB binding sites in their LTRs, leading to higher transcriptional activity in vitro and higher viral loads in people living with HIV-1. In fact, reports have shown the appearance of a fourth, novel NFκB variant in HIV-1C viruses in India [7,8]. HIV-1 Gag p6 contains two late motifs—PTAP and LYPX—which recruit the cellular endosomal sorting complexes required for transport (ESCRT) machinery protein TSG101 or the ESCRT adaptor protein ALIX, respectively, to the virion assembly sites. TSG101 or ALIX both recruit ESCRT III proteins termed charged multivesicular body (CHMP) proteins, which bring about membrane fission at the bud neck and facilitate virus release [9]. Duplication of the PTAP motif is rarely observed in subtype B viruses in infected individuals [10]. In contrast, in HIV-1 subtype C-infected individuals, PTAP duplication is seen in up to 30% of infected individuals [11]. Additionally, a functional LYPX motif is absent in nearly all HIV-1 subtype C HIV-1 due to an LY dipeptide deletion [12,13]. However, it has been reported that in some antiretroviral treatment failure cases, variant viruses emerge containing a PYxE tetrapeptide sequence [14]. Such acquired PYxE insertions restore binding to ALIX, increase virus fitness, and restore increased virus budding in response to CCL2 signaling [12]. Furthermore, while all other HIV-1 subtypes retain a highly conserved dicysteine motif in the Tat protein that is part of a conserved feature between HIV-1 Tat proteins and β-chemokines, HIV -1 subtype C lacks this feature due to a C31S substitution in the dicysteine motif [15].

Genetically engineered reporter HIV-1 constructs have allowed researchers to monitor the kinetics of infection and replication in vitro. Over the years, a large number of recombinant reporter viruses have been designed, developed, and used as tools to study various aspects of HIV-1 replication. Interestingly, a vast majority of these vectors have been based on HIV-1 subtype B. Reporter viruses carrying the *luciferase* gene are used to monitor virus-cell membrane fusion [16], nuclear entry, loss of capsid integrity, and uncoating [17]. Reporter viruses carrying luciferase or green fluorescent protein (GFP) have been used to determine the infectivity of HIV-1 variants, to test their sensitivity to antiretrovirals, to study viral tropism, or to understand the immunological responses to infection. Reporter viruses with *Nef* substitutions are generally used to study early events, while Gag-fusion reporter vectors are designed to study both early and late events of virus assembly and release [18]. Reporter viruses containing fluorescent proteins fused to Vpr or envelope proteins have also been generated to visualize and measure virion production and release. The Vpr-GFP fusion protein helps monitor HIV-1 virions released via fluorescence microscopy or flow cytometry, allowing real-time tracking of viral release and high-throughput screening for factors influencing viral release [19]. The use of Gaussia luciferase fusion with HIV-1 envelope protein allows detection of low levels of virion release. Similarly, NanoLuc-containing bioluminescent HIV-1 reporter viruses facilitated high sensitivity measurement of virus particle production. Kirui and Freed [20] showed how the use of this new reporter virus enhances the efficacy of quantification. More recently, dual reporter viruses have been developed to separately monitor the expression of LTR-dependent and LTR-independent reporter genes to monitor the time to latency from the time of infection [21]. The HIV-1 reporter viruses mentioned above are only a fraction of a larger number that have been reported over the years and that are in use currently. It is unusual that, currently, there is only one reporter virus based on HIV-1 subtype C [22] despite the fact that half of the global HIV burden is due to this subtype of HIV-1. Reporter viruses exemplified by the ones mentioned above have helped vastly advance our knowledge of the biology and pathogenesis of HIV-1 subtype B. A similar progress with subtype C viruses is much needed.

In this study, we generated a mGreenLantern-tagged full-length and replication-competent HIV-1_IndieC1_ reporter virus clone [23] for studies in myeloid cells. First, we established that the infectivity and replication competence in macrophages of HIV-1_IndieC1-mGreenLantern_ (HIV-1_IndieC1-mGL_) virus was similar to that of the parental virus. Second, we investigated whether the new reporter virus retains a characteristic feature of HIV-1 subtype C viruses, namely an inability to respond to CCL2-mediated modulation of virus production, which is due to the characteristic absence of a functional LYPX motif in the Gag p6 late domain of subtype C HIV-1 isolates. We previously showed that CCL2 treatment stimulates the mobilization of ALIX from the F-actin cytoskeleton to the cytoplasm, making it possible for HIV-1 Gag to recruit ALIX efficiently and enhance budding [12]. Due to the lack of LYPX motif on subtype C HIV-1 Gag protein, subtype C viruses are unable to respond to CCL2 in the media to enhance virus production via increased budding. It has also been shown that blocking CCL2-signaling via the addition of anti-CCL2, which leads to a near complete colocalization of ALIX with F-actin structures, causes a strong inhibition of virus production (6–10 fold). We tested the new reporter virus for its ability to respond to CCL2 or anti-CCL2 treatment and found that the virus production is not modulated by the levels of CCL2 present in the medium. Finally, we tested the responsiveness of the new HIV-1 subtype C reporter to methamphetamine (Meth). We report that HIV-1 subtype C virus replication is enhanced by Meth, and the increase is to a much higher level than that observed in HIV-1 subtype B.

## 2. Materials and Methods

### 2.1. Cells, Cell Lines, and Infectious Clones of HIV-1

The 293T cells were obtained from ATCC and TZM-bl cells from the NIH AIDS Reagent Repository. Both cell types were maintained in Dulbecco’s Modified Eagle Medium (DMEM) (Gibco, Thermofisher Scientific, New York, NY, USA) supplemented with 10% FBS (Gibco, Thermofisher Scientific, NY, USA) and 1% Penicillin-Streptomycin (Gibco, Thermofisher Scientific, NY, USA) and incubated at 37 °C with 5% CO_2_. Human monocyte-derived macrophages (MDMs) were obtained by differentiating CD14^+^ monocytes isolated from peripheral blood mononuclear cells (PBMCs) by negative selection (MojoSort Human Pan Monocyte isolation kit, catalog number, 480060, Thermofisher Scientific, NY, USA). Negative selection protocol utilizes beads carrying antibodies to cell surface markers on blood cell types that are not needed, such as T and B lymphocytes, and granulocytes such as neutrophils, eosinophils, basophils, and platelets. Thus, the protocol yields only CD14+ CD16- monocytes. PBMCs were purified from blood samples from the New York Blood Center.

### 2.2. Plasmids

pIndieC1 was a gift from U. Ranga, pNLAD8-GFPNef was a gift from J. Karn, and pADA was a gift from K. Peden. The pcDNA 3.1-mGreenLantern and pEF1a-IRES-Neo were from Addgene.

### 2.3. Construction of HIV-1_IndieC1-mGL_ Reporter Virus

The mGreenLantern-IRES cassette was inserted into the backbone of pIndieC1 by overlap extension PCR [24] as follows. Fragment 1 (2614 bp) represents a part of gp160 starting from *Pac I* restriction site, fragment 2 represents mGreenLantern (720 bp) from pcDNA3.1-mGreenLantern [25], fragment 3 represents IRES (553 bp) derived from pEF1a-IRES-Neo [26], fragment 4 represents *Nef* (624 bp), and fragment 5 represents the rest of the pIndieC backbone (3575 bp), which ends at the *AatII* restriction site. Using repeat sequences that were built-in at the joining ends, fragments I and II were combined by overlap extension PCR, followed by sequentially adding fragments III, IV, and V all via successive overlap extension PCR reactions to the 3′ end of the product of the previous overlap extension PCR step. All PCR reactions were performed using Q5 High-fidelity DNA polymerase (New England Biolabs). Primers used for PCR reactions are listed in Table 1. The final 8 kb product (*Pac1*-gp160-mGreenLantern-IRES-Nef-LTR-*AatII*) was digested with *Pac I* and *Aat II* and ligated to the 6.7 kb *Pac I*-*AatII* fragment derived from parental pIndieC1 plasmid to recreate the entire virus clone with the mGreenLantern reporter gene (Figure 1). The presence of mGreenLantern-IRES in the final pIndieC1-mGreenLantern plasmid and the absence of undesired mutations were validated via nanopore (Plasmidsaurus, Eugene, OR, USA) and Sanger sequencing.

### 2.4. Generating High-Titer Virus Stocks and Testing Their Infectivity

CD14^+^ monocytes were plated with Monocyte attachment medium (PromoCell, Heidelberg, Germany) and differentiated into macrophages by adding DMEM supplemented with 10% FBS, 1% Penicillin-Streptomycin, and recombinant human macrophage colony stimulating factor (rhM-CSF; 5 ng/mL, Peprotech, Themo Fisher, Rocky Hill, NJ, USA, Catalog number, AF-300–25) and then grown in a cell culture incubator for ten days.

To generate high-titer viruses, plasmid DNAs of infectious molecular clones HIV-1_ADA_ (subtype B) [27], HIV-1_IndieC1_ (subtype C) [23], HIV-1_NLAD8-GFP-Nef_ [28], and HIV-1_IndieC1-mGL_ reporter were transfected separately (20 µg each) into 293T cells seeded at a density of 10 × 10^6^ cells in a 10 cm plate using Lipofectamine 3000 (Invitrogen, ThermoFisher, Waltham, MA, USA). Media containing infectious viruses were collected at 24 h post transfection for p24 measurement. The p24 levels were measured using an AlphaLisa detection kit (Perkin Elmer, Waltham, MA, USA, cat #AL291F) for HIV-1_ADA_ and HIV-1_NLAD8-GFP-Nef_ (subtype B) or using the HIV-1 p24 Antigen Capture Assay kit (ABL, Rockville, MD, USA; Catalog number, 5421) for HIV-1_IndieC1_ and HIV-1_IndieC1-mGL_ (subtype C). The infectivity of viruses thus generated was tested using TZM-bl luciferase assay. Briefly, TZM-bl cells were seeded in 12-well plates at a density of 2 × 10^5^ cells per well and incubated at 37 °C with 5% CO_2_ for 24 h. Cells were infected with each of the four HIV-1 viruses mentioned above at increasing multiplicities of infection (MOI). In this study, we define MOI as ng of p24 of HIV per 2 × 10^5^ target cells. We used MOIs equivalent to 31.25, 62.5, 125, 250, and 500 ng p24 per 2 × 10^5^ cells and incubated them at 37 °C with 5% CO_2_ for 24 h. Cells were washed twice with 1xXPBS and lysed using lysis buffer in the Luciferase assay kit (Promega, Madison, WI, USA). For each virus, 50 μL of cell lysate each were used to measure the luciferase activity.

### 2.5. Multiday HIV-1 Replication Assay in Monocyte-Derived Macrophages

Macrophage infection and replication of HIV-1_ADA_, HIV-_1NLAD8-GFP-Nef_ reporter, HIV-1_IndieC1_, and HIV-1_IndieC1-mGL_ reporter viruses were tested in monocyte-derived macrophages (MDMs). Briefly, CD14^+^ monocytes were seeded in 6-well plates at a density of 1 × 10^6^ cells per well and differentiated into macrophages, as described above. MDMs were infected with each of the four different viruses separately at 10 ng p24 per 10^6^ cells. Half of the media (1 mL) was removed and replenished with fresh medium every three days throughout the multiday replication experiment. Replication competence was determined by measuring p24 in cultured infected media. All MDM experiments were performed three times. Each experiment used MDMs from a separate donor, and two experimental replicates were used to calculate an average per donor. Thus, the data in each case are represented as mean ± SEM from three separate samples (*n* = 3).

### 2.6. Measuring the Effect of CCL2 and Anti-CCL2 on the Replication of HIV-1 Isolates

To determine the effect of CCL2 and anti-CCL2 on the replication of HIV-1, 1 × 10^6^ MDMs were infected with HIV-_1NLAD8-GFP-Nef_ and HIV-1_IndieC1-mGL_ reporter viruses at 10 ng p24 per 10^6^ cells. Infected MDMs were treated with CCL2 (250 ng/mL; PeproTech, Themo Fisher, Rocky Hill, NJ, USA, Catalog number, 300-04-20UG), anti-CCL2 (2.5 mg/mL; PeproTech, Themo Fisher, Rocky Hill, NJ, USA, Catalog number, 500-M71-500UG), or an isotype control antibody from unimmunized mice (IgG2a, kappa; 2.5 mg/mL; PeproTech, Themo Fisher, Rocky Hill, NJ, USA, Catalog number, 500-M00) [12]. Half the media and all the treatments were replenished every 3 days for 30 days. Replication of HIV-_1NLAD8-GFP-Nef_ and HIV-1_IndieC1-mGL_ reporter viruses was monitored by tracking the levels of cellular green fluorescence using an inverted fluorescence microscope (Zeiss Axio Observer, Carl Zeiss, Oberkochen, Germany) or by measuring p24 levels in the cultured infected media as mentioned above.

### 2.7. Measuring the Effect of Methamphetamine on the Replication of HIV-1 Isolates

To investigate the effect of Meth on the HIV-1 (subtype B and subtype C) replication, MDMs were seeded in 6-well plates at a density of 1 × 10^6^ cells/well and infected at day 10 with HIV-1_ADA_, HIV-1_NLAD8-GFP-Nef_, HIV-1_IndieC1_, or HIV-1_IndieC1-mGL_ viruses, respectively, at 10 ng p24/10^6^ cells. Infected MDMs were treated with Meth (Sigma-Aldrich, St. Louis, MO, USA, catalog number, 51-57-0) at three different concentrations (100, 250, and 500 μM). Every three days, half the media was replenished with new media containing fresh Meth for 15 days. The effect of Meth on viral replication was determined by monitoring p24 production in the media for all infected viruses as well as by tracking the intensity green fluorescence (Zeiss Axio Observer, Carl Zeiss, Oberkochen, Germany) for HIV-1_IndieC1-mGL_ and HIV-_1NLAD8-GFP-Nef_ viruses.

### 2.8. Quantification of Fluorescence Intensity

Cellular green fluorescence resulting from the expression of fluorescent proteins (GFP or mGreenLantern) upon HIV infection was quantified using a Zeiss Axio Observer epifluorescence microscope at magnification (20×). Images were captured at the center of the well, with an average of 50 cells imaged for each reporter virus. We used NIH Image J software (NIH Image Manual (V1.61)) to quantify the fluorescence intensity image. Intensities for each time point were calculated from images taken from three replicates and plotted in respectively.

### 2.9. Statistical Analysis

At least three independent replicates were carried out for each experiment, and the mean was calculated in all cases. Statistical analyses were performed using GraphPad Prism 8.0 (GraphPad Software, San Diego, CA, USA).

## 3. Results

### 3.1. Construction of HIV-1_IndieC1mGL_ Reporter Virus

We inserted the sequences encoding mGreenLantern at the beginning of the *Nef* gene of an infectious molecular clone of HIV-1_IndieC1_ such that the ATG at the start of *Nef* open reading frame (orf) would be the initiation codon for mGreenLantern. An internal ribosomal entry site (IRES) sequence was inserted upstream of *Nef* gene while restoring its ATG site to ensure that *Nef* expression would be unaffected. No additional modifications were made to HIV-1_IndieC1_. Figure 1 shows the position of mGreenLantern insertion and the IRES sequence inserted in the HIV-1_IndieC1_ construct in comparison with the parental HIV-1_IndieC1_ clone. The reporter construct was sequenced via both nanopore sequencing of the whole viral construct and Sanger sequencing to resolve any ambiguities. The sequencing confirmed that the construct included the intended modifications and no undesired mutations.

### 3.2. HIV-1 _IndieC1-mGL_ Reporter Virus Is a Replication Competent Virus

We wished to test the infectivity and replication competence of the new reporter virus HIV-1_IndieC1-mGL_. We prepared high-titer virus stocks of HIV-1_IndieC1-mGL_ and its parental HIV-1_IndieC1_ virus. For comparison, we also generated virus stocks of a subtype B reporter virus, HIV-1_NLAD8-GFP-Nef_, and its parental virus HIV-1_ADA_. First, we tested the infectivity of these four viruses employing the TZM-bl luciferase assay. TZM-bl cells carry an integrated copy of the luciferase reporter gene under the control of HIV-1 LTR [29] and thus respond with the induction of luciferase activity that corresponds to the degree of infection. TZM-bl cells were infected with increasing virus inputs for each of the four viruses. The results indicated that the infectivity of the HIV-1_IndieC1-mGL_ reporter virus was comparable to its parental virus HIV-1_IndieC1_ (Figure 2A). In our comparison of the infectivity of the subtype B virus (HIV-1_ADA_) and the subtype B reporter virus, HIV-1_NLAD8-GFP-Nef_, the reporter virus was consistently higher than that of HIV-1_ADA_ by about 2-fold at all multiplicities of infection (Figure 2B). This is likely due to the chimeric derivation of this virus [28]. The HIV-1_NLAD8-GFP-Nef_ virus is based on HIV-1_NL4-3_ provirus but replaces the *env* gene with that from HIV-1_ADA_. As the only common feature between HIV-1_ADA_ and HIV-1_NLAD8-GFP-Nef_ is the *env* gene of HIV-1_ADA_, the robustness of the virus may be attributable to sequences outside *env*.

Next, we examined the replication competence of the HIV-1 subtype C reporter virus alongside control viruses (HIV-1_IndieC1_, HIV-1_ADA_, and HIV-1_NLAD8-GFP-Nef_) in MDMs in a 30-day replication assay. We observed a delay in virus replication for HIV-1_IndieC1mGL_ when compared to the parental HIV-1_IndieC1_ virus (day 18 peak p24 production vs. day 15; Figure 3A left panel). This is not surprising considering the two macro changes made to the virus genome, which may have affected some aspect of the genome RNA or any of the downstream functions of the genome, with a consequent effect on the virus progeny population that manifested in a 3-day delay. It is well known that HIV-1 subtype C viruses replicate at a lower efficiency than subtype B viruses. Accordingly, we found that both HIV-1_ADA_ and HIV-1_NLAD8-GFP-Nef_ viruses replicated at a higher rate, reaching a peak virus production that was about 3-fold higher than subtype C viruses (compare Figure 3A,B). Collectively, the results of the p24 to infectivity ratios (relative luminescence units of luciferase activity to ng p24) and the replication-competence experiment show that the HIV-1_IndieC1-mGL_ is a replication-competent virus similar to the parental HIV-1_IndieC1_.

### 3.3. Production of HIV-1_IndieC1-mGL_ in MDMs Is Not Responsive to CCL2 Levels

Previous reports showed that CCL2 immuno-depletion suppresses HIV-1 subtype B replication in macrophages [12,30] and that the addition of CCL2 enhances virus replication [12]. We showed that this CCL2-dependent modulation of HIV-1 replication was due to the mobilization of ALIX from F-actin stress fibers to the soluble cytoplasm by CCL2 signaling, and this required the presence of late domain motif LYPX in Gag p6, which is responsible for ALIX recruitment to virus assembly and budding sites on the plasma membrane. Furthermore, we reported that 99.8% of the 495 full-length subtype C virus sequences tested lacked a functional LYPX motif due to a highly conserved LY dipeptide deletion in Gag p6. Thus, the subtype C virus HIV-1_IndieC1_ was unresponsive to levels of CCL2 in the media due to its inability to recruit ALIX [12]. Since the efficiency of the budding and release of virus particles is not responsive to the levels of CCL2 in HIV-1 subtype C viruses, we tested the HIV-1_IndieC1-mGL_ reporter virus for its CCL2 responsiveness. We infected MDMs with either HIV-1_IndieC1-mGL_ or the parental HIV-1_IndieC1_ virus. For each virus, four separate conditions were employed, namely untreated, CCL2 treated, anti-CCL2 treated, or isotype control IgG-treated, and the virus particle release was measured over a period of 30 days. Like before, we observed a 3-day delay in the replication of HIV-1_IndieC1mGL_ virus (Figure 4A, Left panal).

MDMs infected with either HIV-1_NLAD8-GFP-Nef_ or HIV-1_IndieC1-mGL_ virus displayed an increase (virus production peak on 18 days instead of 15 days post infection). We also included a subtype B control HIV-1_NLAD8-GFP-Nef_, which showed virus production at 15 days post infection. Importantly, anti-CCL2 treatment resulted in a substantial reduction (~6-fold) in the production of subtype B virus compared to untreated infected MDMs or isotype antibody control (Figure 4A, right panel). Similarly, CCL2 treatment enhanced the production of subtype B viruses by ~1.8-fold compared to untreated or isotype antibody-treated MDMs (Figure 4A, right panel). In contrast, virus production levels for HIV-1_IndieC1mGL_, as expected, were unaffected by levels of CCL2 in the medium (Figure 4A, left panel).

In addition to monitoring p24 production, we also quantified mGreenLantern or GFP fluorescence, respectively, for cells infected with HIV-1_IndieC1-mGL_ and HIV-1_NLAD8-GFP-Nef_ viruses, respectively. A gradual increase was observed in the total fluorescence intensity from the time of infection to day 15 post infection. For HIV-1_NLAD8-GFP-Nef_, treatment with anti-CCL2 led to a considerable reduction in GFP fluorescence compared to untreated or isotype antibody-treated, infected MDMs (Figure 4C). Similarly, treatment with CCL2 led to an increase in GFP fluorescence throughout the culture up to day 15 (Figure 4C), as reported previously [12]. In contrast, addition of CCL2, anti-CCL2, or the isotype control antibody did not change the fluorescence intensity of HIV-1_IndieC1-mGL_-infected MDMs (Figure 4B). A fluorescence intensity plot for each treatment (e.g., +CCL2, +anti-CCL2, or +Isotype antibody control) for infection with each reporter virus (HIV-1_IndieC1-mGL_ or HIV-1_NLAD8-GFP-Nef_) showed results consistent with the observation that the subtype C reporter is refractory to CCL2 levels, unlike the subtype B reporter virus (Figure 4D, compare left and right panels). Furthermore, the pattern of responsiveness to CCL2 levels by the two reporter viruses (Figure 4D) was similar to that seen when the p24 production was plotted (Figure 4A). These results together indicate that the replication pattern of the newly created HIV-1_IndieC1-mGL_ follows the patten of the wildtype HIV-1_IndieC1_ and that the replication of this virus can be monitored by measuring p24 as well as by imaging and by quantitating mGreenLantern fluorescence.

### 3.4. Methamphetamine Treatment Markedly Enhanced HIV-1 Subtype C Replication in MDMs

Previous studies have reported that Meth increases HIV-1 replication in macrophages [31,32,33]. Meth is known to trigger the NF-kB signaling pathway in immune cells, including macrophages, and thus, its impact on HIV-1 replication is mediated by enhanced LTR transcription [22]. Since HIV-1 subtype C viruses contain 3 NF-kB sites, we chose to study the impact of Meth on subtype C replication in MDMs, employing the HIV-1_IndieC1-mGL_ reporter virus. We infected MDMs, in parallel, with HIV-1_ADA_, HIV-1_NLAD8-GFP-Nef_, HIV-1_IndieC1_, and HIV-1_IndieC1-mGL_ viruses in the presence or absence of Meth. As indicated by increased levels of p24 measured at various times after infection up to day 15, Meth treatment enhanced the replication of all viruses compared to untreated HIV-infected MDMs (Figure 5). However, the increase was more robust for HIV-1 subtype C viruses compared to subtype B viruses (compare Figure 5A, top panels to bottom panels). Also, we noticed that the highest increase in the replication of HIV-1 (subtype C and B) by Meth was observed at a concentration of 100 μM, while the increases were progressively lower at 250 and 500 μM, respectively. For example, the increase in p24 levels in the presence of Meth over its absence for HIV-1_IndieC-mGL_ was ~4.5-fold at 100 μM, 3.7-fold at 250 μM, and 2.6-fold at 500 μM (Figure 5A, top right). Increases for HIV-1_IndieC1_ were remarkably similar to that observed for HIV-1_IndieC-mGL_ at each of the three Meth concentrations (Figure 5A, top left). The increases in virus replication for HIV-1_ADA_ or HIV-1_NLAD8-GFP-Nef_ were lower than subtype C viruses: 3- or 3.8-fold at 100 μM, 2.2- or 2.7-fold at 250 μM, and 1.4- or 1.7-fold at 500 μM, respectively (Figure 5A, bottom panels).

To determine if fluorescence intensity of the infected cells reflects the observed Meth-induced increase in replication of the viruses, we also compared the results of cellular green fluorescence intensity in MDMs infected with HIV-1_IndieC1-mGL_ or HIV-1_NLAD8-GFP-Nef_ in the presence or absence of Meth at three different concentrations (100, 250, and 500 μM). First, we noted that both viruses seem to display a similar level of fluorescence intensity despite the fact that mGreenLantern is 6-fold brighter than GFP. The reasons for this are that HIV-1_NLAD8-GFP-Nef_ virus replicates much more robustly than HIV-1_IndieC1mGL_ both because subtype B viruses replicate better than subtype C but also because the chimeric nature of HIV-1_NLAD8-GFP-Nef_ virus replicates to higher levels, as mentioned above. We therefore made comparisons (only within each virus not between the viruses) of green fluorescence intensity in the presence of Meth compared to its absence (Figure 5B). The increase in fluorescence intensity observed over no Meth was much higher for HIV-1_IndieC1-mGL_ (Figure 5C, left panel) than for HIV-1_NLAD8-GFP-Nef_ (Figure 5C, right panel). We noted that the fluorescence intensities of GFP and mGreenLantern are intrinsically different. Therefore, the comparisons were made between treatments with and without Meth but within either HIV-1_IndieC1-mGL_ alone or HIV-1_NLAD8-GFP Nef_ alone. These findings strongly suggest that the impact of Meth on HIV-1 subtype C replication is enhanced to a higher level than subtype B due to the additional NF-kB binding site in the LTR of HIV-1_IndieC1_ or HIV-1_IndieC1-mGL_.

## 4. Discussion

In this study, we constructed a HIV-1 subtype C virus expressing mGreenLantern via the spliced *Nef* mRNA. mGreenLantern is a GFP derivative that has been optimized for efficient folding and rapid expression [25]. As a result, mGreenLantern is six times brighter than EGFP. As the reporter virus was based on the macrophage-tropic HIV-1_IndieC1_, this virus would be useful for studying HIV-1 replication in myeloid cells (monocytes, macrophages, and microglia). The design of this reporter virus will facilitate translation of the Nef protein from the primary, unspliced mRNA as well as from all spliced mRNAs due to the IRES inserted downstream of mGreenLantern. This new reporter virus allows one to track HIV-1 replication by monitoring mGreenLantern expression in the target cells. This method provides a straightforward visual indicator of infection, allowing for efficient tracking and analysis of viral replication in vitro. As shown by our results, HIV-1_IndieC1-mGL_ replicated with equal efficiency to the parental HIV-1_IndieC1_ virus—including in the infectivity assay using TZM-bl cells and a multiday replication assay in MDMs (Figure 2 and Figure 3). We note that in spite of displaying infectivity levels that were similar to the parental virus, the reporter virus displayed a delayed peak productivity (3 days). While it is hard to pinpoint the mechanistic basis for this 3-day delay, one could imagine that the inserted sequences may affect the stability of chimeric viral genomic RNA. However, in the one month-long macrophage replication experiments we conducted (Figure 3 and Figure 4), there were no observable defects. Not only did HIV-1_IndieC1-mGL_ replicate with equal efficiency to the parental virus, but it also retained its lack of responsiveness to CCL2, which indicates that it also retains other characteristics of HIV-1_indieC1_. HIV-1_IndieC1-mGL_ macrophage-tropic subtype C reporter virus complements the only other currently available subtype C reporter virus reported by Rai et al., which employed EGFP as a reporter and was based on K3016, a South African subtype C virus that could infect T cells and PBMCs [34]. Thus, HIV-1_IndieC1-mGL_ represents the only HIV-1_IndieC1_-derived virus with a fluorescence reporter.

The presence of three NF-κB binding sites in the LTR region of subtype C HIV-1 compared to only two NF-κB binding sites in most subtypes of HIV-1, including subtype B, suggest that HIV-1 subtype C may respond to Meth with a greater responsiveness than HIV-1 subtype B [35]. Therefore, we examined the effect of Meth treatment on MDMs infected with recombinant HIV-1_IndieC1-mGL_ and HIV-1_NLAD8-GFP-Nef_ viruses. Our results revealed that the Meth treatment of MDMs infected with the HIV-1_IndieC1-mGL_ reporter virus markedly increased the viral replication compared to that of the infected cultures that were left untreated. This increase was also observed in the parental virus HIV-1_IndieC1_. This increase was higher than that observed for HIV-1_NLAD8-GFP-Nef_ compared to similarly infected untreated MDMs. As expected, Meth treatment of MDMs infected with the parental subtype B virus HIV-1_ADA_ was comparable to that of HIV-1_NLAD8-GFP-Nef_.

Meth is known to contribute to the progression of AIDS by enhancing viral replication rates and the consequent ineffectiveness of antiretrovirals [36]. In vitro, Meth increases viral replication in monocytes [33,37] and in CD4 T cells [30], which can lead to increased viral load in HIV-1-positive active drug users. However, it must be emphasized that the increase in HIV-1 replication in CD4 T cells [30] is controversial, as an earlier study showed that Meth inhibits of HIV-1 replication in T cells [38]. Interestingly, both these studies used comparable Meth concentrations in activated CD4 T cells and used the same strain of HIV-1 (HIV-1_BAL_). Bosso et al. showed that the increase in HIV-1 replication observed in the presence of Meth is mediated by NF-kB enhancers in HIV-1 subtype B [35].

Our results also indicate a dose-dependent response of Meth in inducing HIV-1 replication. We found that lower concentrations of Meth increase virus replication better. Previous reports have shown increases in HIV-1 replication upon Meth addition. One report studied the effect of Meth on the HIV-infected macrophages at a range of concentrations (1 µM to 250 μM) [31]. Quantitation of HIV-1 RNA revealed that the highest increase in HIV by Meth was at 250 μM at 8 days post infection, an approximately two-fold increase over untreated, infected macrophages. In our experiments for HIV-1 subtype B virus, the highest increases in virus replication were observed at 100µM Meth (up to 3-fold higher) at 15 days post infection. At the higher concentrations of Meth tested (250 µM and 500 µm), the extent of inhibition was reduced to 2-fold or less. We do not know the basis for this type of dose response. Another study investigated the effect of Meth on the expression of the HIV genome in human microglia in vitro [33]. They found that Meth treatment increased the transcriptional activity of the HIV-1 LTR promoter in a dose-dependent manner. In that study, Meth increased the activity and movement of the transcription factor NF-κB, which is a key player in transcriptional activation from HIV-1 LTR. Blocking NF-κB signaling prevented the Meth-induced activation of HIV-1 LTR. These findings suggest that Meth can directly stimulate HIV gene expression in microglial cells, the primary target cells for HIV-1 in the central nervous system, through NF-κB activation. These results support our finding of a much higher increase in the stimulation of viral replication at 100 µM Meth in the presence of 3 NF-κB sites in the LTR as in HIV-1 subtype C HIV-1.

Our mGreenLantern-tagged HIV-1_IndieC1_ reporter virus can be beneficial for in vitro studies involving monocytes, microglia, and macrophages. Although we did not perform an exhaustive series of tests, minimally, the reporter virus retained the replication characteristics of its parental clone that were tested, which indicates that it can be used for cell culture and possibly in small animal models such as humanized mice. Using this reporter virus, we were able to detect the effects of Meth on HIV-1 replication of both subtype C and B in infected MDMs, where the responses to Meth of HIV-1 subtype C were higher than that of HIV-1 subtype B.

## 5. Conclusions

We have shown that a mGreenLantern-tagged subtype C virus, that we constructed, is replication-competent and retains some of the characteristic features of subtype C that we tested and that its responsiveness to Meth treatment is more robust than subtype B. The infectious full-length recombinant mGreenLantern-tagged HIV-1_IndieC1_ reporter virus can be an essential tool for understanding the intracellular pathways inside the cells that control the virus replication. Furthermore, it can be used for conducting HIV subtype C-specific research and investigating the viral and host factors influencing the rapidly expanding subtype C infections worldwide.

## Figures and Tables

**Figure 1 viruses-16-01859-f001:**
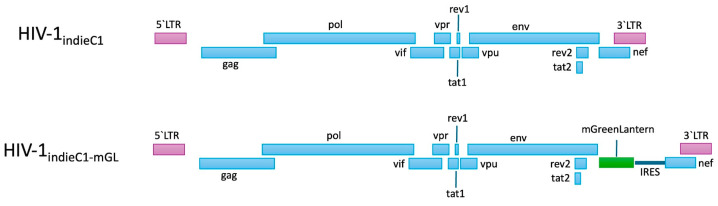
Schematic representation of the mGreenLantern expressing HIV-1 subtype C reporter virus in comparison with the parental HIV-1_IndieC1 virus_. All of the open-reading frames (*gag* (1476 nt), *pol* (3000 nt), *vif* (579 nt), *rev1* (75 nt), *tat1* (216 nt), *vpu* (249 nt), *vpr* (291 nt), *env* (2574 nt), *tat2* (90 nt), *rev2* (228 nt), *mGreenLantern* (720 nt), and *Nef* (624 nt)) are shown as boxes and labeled as such. An IRES sequence (shown as a horizontal thick line upstream of nef) was inserted at5′ of the *Nef* gene to facilitate its expression.

**Figure 2 viruses-16-01859-f002:**
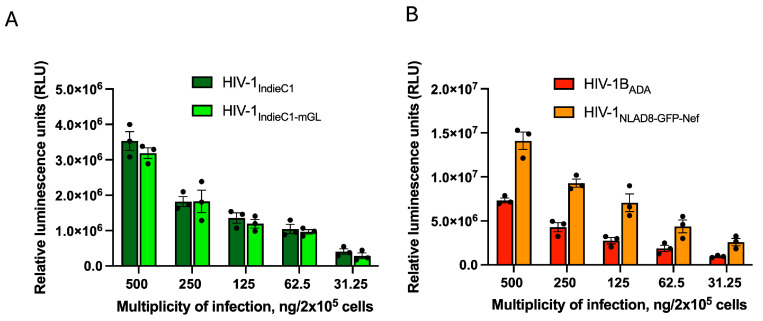
Infectivity of HIV-1 subtype C and B viruses measured by luciferase assay in TZM-bl cells. TZM-bl cells were infected with HIV-1 subtype C and B with different MOI (500, 250, 125, 62.5, and 31.25 ng p24/2 × 10^5^ cells) and incubated at 37 °C with 5% CO_2_ for 24 h. (**A**) Infectivity of recombinant HIV-1_IndieC1-mGL_ reporter virus measured alongside the parental HIV-1_IndieC1_. (**B**) Infectivity of recombinant HIV-_1NLAD8-GFP-Nef_ measured alongside HIV-1_ADA_. Data are represented as mean ± SEM (*n* = 3).

**Figure 3 viruses-16-01859-f003:**
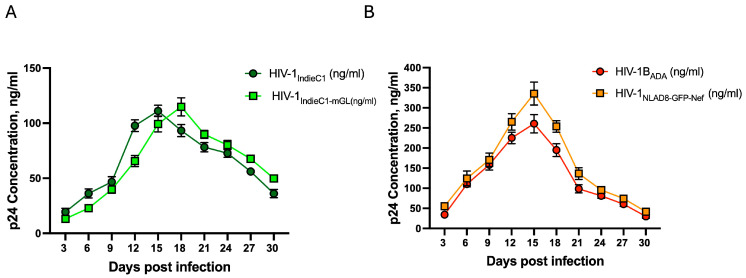
Measurement of replication efficiency of HIV-1 (subtype B and subtype C) using a multi-day replication assay in MDMs. MDMs (10 ng/1 × 10^6^) were infected with either subtype C or subtype B viruses. (**A**) The replication competence of recombinant HIV-1_IndieC1-mGL_ reporter virus was compared to the parental virus, HIV-1_IndieC1_. (**B**) The replication competence of recombinant HIV-1_NLAD8-GFP-Nef was_ compared to HIV-1_ADA_. Data are a mean of three experiments ± SEM (*n* = 3).

**Figure 4 viruses-16-01859-f004:**
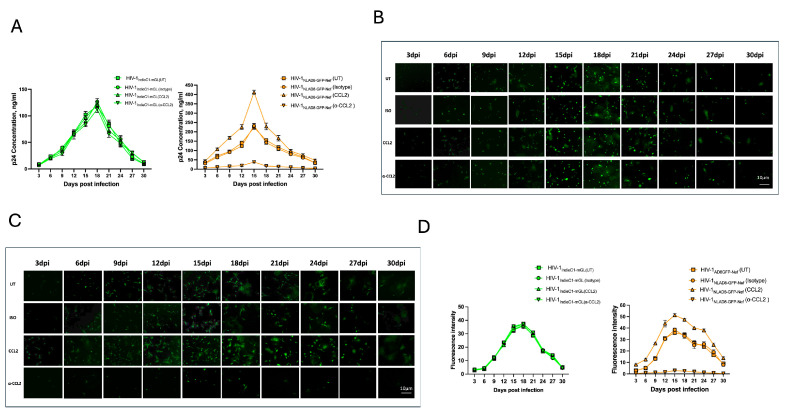
Effect of CCL2 on HIV-1 subtype C and B replication in MDMs. (**A**) MDMs were infected with HIV-1_IndieC1-mGL_ reporter virus (**Left**) or HIV-1_NLAD8-GFP-Nef_ (**Right**) and treated with CCL2, anti-CCL2, or isotype control antibody (see Methods). UT = no treatment. (**B**) Tracking the effect of CCL2 or anti-CCL2 on HIV-1_IndieC1-mGL_ reporter replication via fluorescence intensity. (**C**) Tracking the effect of CCL2 or anti-CCL2 on HIV-1_NLAD8-GFP-Nef_ replication in MDMs using fluorescence intensity. dpi, days post infection. Scale bar for (**B**,**C**) is as in the panel anti-CCL2, 30 dpi. (**D**) Measurement of fluorescence intensity in panels (**B**,**C**) for subtype C or subtype B reporter viruses with various treatments. Data are a mean of three experiments ± SEM (*n* = 3).

**Figure 5 viruses-16-01859-f005:**
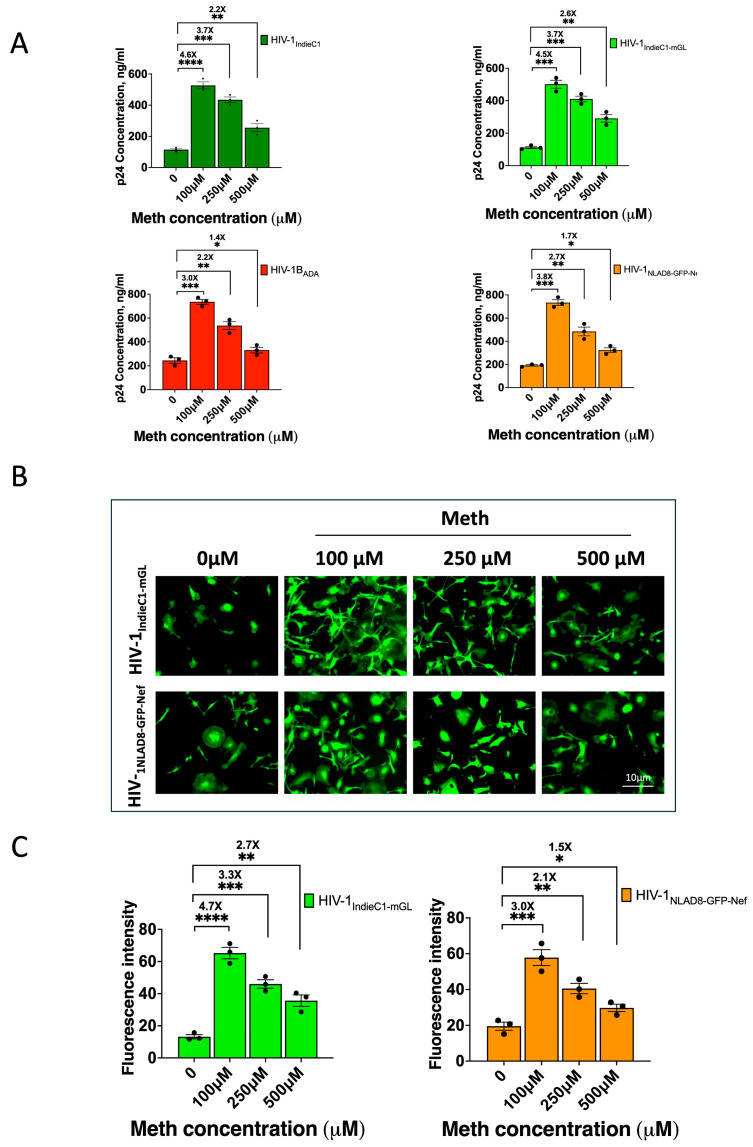
Effect of Meth on HIV-1 replication. (**A**) Differential effect of Meth on HIV-1 subtype C (**top panels**) and subtype B (**bottom panels**) replication in MDMs. MDMs (10 ng/1 × 10^6^) were infected with subtype C or B viruses and treated with Meth at 100, 250, or 500 μM or not treated (0 µM). Replication of HIV-1_IndieC1_ (**top left**); HIV-1_IndieC1-mGL_ reporter (**top right**); HIV-1_ADA_ (**bottom left**) and HIV-1_NLAD8-GFP-Nef_ (**bottom right**) in the presence or absence of Meth as measured via p24 levels at 15 dpi. *p*: *, 0.05; **, <0.01; ***, <0.001; and ****, <0.0001. (**B**) Tracking the effect of Meth on HIV-1_IndieC1-mGL_ or HIV-1_NLAD8-GFP-Nef_ via green fluorescence intensity. (**C**) Measurement of fluorescence intensity to determine effect of Meth on HIV-1 replication from Panel B. Shows plots of fluorescence intensity for HIV-1_IndieC1-mGL_ (**left**) and HIV-1_NLAD8-GFP-Nef_ (**right**). Data are a mean of three experiments ± SEM (*n* = 3). Scale bar = 10 µm. *p*: *, 0.05; **, <0.01; ***, <0.001; and ****, <0.0001.

**Table 1 viruses-16-01859-t001:** List of primers used in overlap PCR to build the *Pac1*-gp160-mGreenLantern-IRES-Nef-LTR-*AatII* fragment.

Item	Name of Primer	Sequence of Primer
1	Fragment 1 (Forward)	5′GGTTAATTAAAAGAATTAGGGAAAGAGCAG 3′
2	Fragment 1 (Reverse)	5′CCCTTGCTCACCATTATTATTTTATTGCAAAGCTGCTTCAAAGC 3′
3	Fragment 2 (Forward)	5′GCTTTGCAATAAAATAATAATGGTGAGCAAGGGCGAGGAGC 3′
4	Fragment 2 (Reverse)	5′AGCGGCTTCGGCCAGTAACGTTTACTTGTACAGCTCGTCC 3′
5	Fragment 3 (Forward)	5′TGGACGAGCTGTACAAGTAAACGTTACTGGCCGAAGCCGC 3′
6	Fragment 3 (Reverse)	5′TTTGACCACTTGCCCCCCATTATTATCATCGTGTTTTTCAAAG3′
7	Fragment 4 (Forward)	5′TTGAAAAACACGATGATAATAATGGGGGGCAAGTGGTCAAAATG3′
8	Fragment 4 (Reverse)	5′CGGAAAGTCCCTTCTGTGTCAGCAGTCTTTGTAAAACTCCGG3′
9	Fragment 5 (Forward)	5′CCGGAGTTTTACAAAGACTGCTGACACAGAAGGGACTTTCCG3′
10	Fragment 5 (Reverse)	5′GGTGGTGACGTCAGGTGGCACTTTTCGGGG 3′

## Data Availability

Following publication, the authors are willing to share all raw data that have been analyzed in the manuscript, and they will be made available to other researchers.

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
