# Peer review of "Construction of a Macrophage-Tropic Subtype C HIV-1-mGreenLantern Reporter Virus for Studies on HIV-1 Replication and the Impact of Methamphetamine"

_viruses, 2024, doi:10.3390/v16121859_

Round 1
Reviewer 1 Report
Comments and Suggestions for Authors
The authors constructed and tested replication of a fluorescent protein-expressing HIV subtype C variant by introducing mGreen-Lantern-IRES sequences between the Env and Nef encoding regions. Several other HIV reporter viruses have been published, but most of them are based on subtype B strains. This new virus expands the toolbox for in vitro (and possibly mice) studies focusing on HIV subtype C, which is responsible for a large part of the global HIV infections. The paper is straight-forward and the results are nice and sound. I believe that the text and figures can be improved, as indicated below.
Line 121-124 these methods should be described in more detail or references should be indicated where these details can be found (for example, what is “negative selection” and “leukopaks”?)
Line 131-…. What is meant with overlap PCR? Can the authors give a reference or a bit more details?
Line 151: Where does rhM-CSF stand for?
Line 183-184: please show the origin (supplier/catalogue number) of the CCL2, anti-CCL2 and isotype control antibody. What is the isotype?
Line 194: Explain abbreviation Meth the first time when metamphetamine is mentioned. please show the origin (supplier/catalogue number).
Line 212: The authors should start the results section with the construction of the new virus, as this is their first result. Figure 1 can be used to compare the original virus with the green-lantern variant.
Figure 1 should be improved anyway, as the boxes representing the different open-reading frames are not correctly sized and positioned (most serious error: Rev 2 overlapping with green lantern instead of Tat2). All boxes should have a size reflecting the size of the orf, and they should be correctly positioned.
Line 232: comparable? replication of the green-lantern virus is clearly slightly delayed compared to the original virus (3-days delay). This is no surprise and the authors could discuss probable causes (for example in the discussion).
Line 237: what is meant with “the p24 to infectivity ratios”? Do the authors mean “the infectivity experiments?
Line 262: via = measured by?
Line 263: the use of the term MOI/multiplicity of infection is not correct. It is the amount of virus based on ng CA-p24 that was varied as input.
Line 265-266: remove “was comparable”. Results/interpretations should be mentioned in the text and not in the legend. Moreover, the 2 viruses in B do NOT have a comparable infectivity.
Line 283-284: 3 times, 3 separate donors, 2 replicates…… how does this result in n=3?
Line 302: as shown in our results >>> as shown in Figure… or remove
Line 309: we monitored replication upon infection of MDM….
Line 319: ….. for each treatment for infection with…… this sentence is not clear
Line 326: …. and by following the fluorescence pattern? Do the authors mean by quantification of the fluorescence?
Figures 4 and 5 are too small and can only be seen when enlarged on screen and not in a print.
Line 343. Data are represented….. Should this sentence be moved to the end of the legend? 3 donors, 2 replicates, how do the authors get to n=3?
Line 355: in the presence of absence of Meth (Figure….). As indicated by the increased (?) levels of P24….
Line 363: 3.65 and 2.58 suggest very exact values…. 3.7 and 2.6 are more in line with 4.5. Same situation in line 367.
Line 384: remove differential
Line 385 and other positions in legend: subtype C is shown in the bottom panels and not in the top panels!
Line 390 and 395: P: *, ,0.05? Also here explain n=3
The authors should mention/discuss that, in their new virus, Nef is (or can be) produced from all spliced and unspliced viral transcripts (translation initiating at the IRES), and not only from the nef RNAs, which could result in an aberrant Nef production (see for example analysis of IRES-Nef constructs in Van der Velden et al. http://dx.doi.org/10.1016/j.virol.2015.11.004).
The authors could in the discussion also mention/discuss the stability of the introduced sequences in the virus (green lantern and IRES sequences will probably be lost upon long term culturing of the virus, but no problem in many short-term assays, like the assays shown in the paper).
Line 458: the authors state that the reporter virus has all the replication characteristics of its parental clone. That is true for the characteristics that they tested in the specific assays, but based on these assays one cannot say for all replication characteristics (as the introduced sequences will likely have all kind of not-identified effects, like an effect on Nef production, viral RNA splicing and possibly other processes).
Author Response
Comment 1 : Line 121-124 these methods should be described in more detail or references should be indicated where these details can be found (for example, what is “negative selection” and “leukopaks”?)
Response: We have now included in the revised manuscript, the following information: “Negative selection protocol, which utilizes beads carrying antibodies to cell surface markers on other blood cell types such as lymphocytes (T and B), granulocytes (e.g., neutrophils, eosinophils, basophils) and platelets, remove all cells other than CD14+ CD16- monocytes.” Line : 123.
We understand that all readers may not know what is a leukopak. Therefore, we have simplified the expression as follows: “PBMCs were purified from blood samples from the New York Blood Center. Line: 126
-Comment 2 : Line 131-…. What is meant with overlap PCR? Can the authors give a reference or a bit more details?
Response: As this is a standard molecular biology protocol, we did not provide additional details for this term. We agree that some readers may be unfamiliar with it. Therefore, the following information has now been added to the revised manuscript: “Overlap PCR or overlap extension PCR is a modification of standard PCR used to assemble two smaller double stranded DNA fragments into a single, longer DNA fragment.” Also a relevant citation has been provided. see line : 134.
-Comment 3 : Line 151: Where does rhM-CSF stand for?
Response: This stands for rhM-CSF. We have added the following explanation: “Recombinant human macrophage colony stimulating factor (rhM-CSF).” See line : 153-154.
-Comment 4 : Line 183-184: please show the origin (supplier/catalogue number) of the CCL2, anti-CCL2 and isotype control antibody. What is the isotype?
Response: The same IgG isotype as the ant-CCL2 antibodies used (in this case IgG2a, kappa). Now mentioned along with the sources and catalog numbers. Line : 188-191.
-Comment 5 : Line 194: Explain abbreviation Meth the first time when metamphetamine is mentioned. please show the origin (supplier/catalogue number).
Response : Kindly note that Line 194 was not the first mention of Methamphetamine. It was first mentioned on line 111, where we have indeed indicated the abbreviation in parentheses. The source of Meth and catalog number are now mentioned under Materials and Methods. Line : 200.
-Comment 6 : Line 212: The authors should start the results section with the construction of the new virus, as this is their first result. Figure 1 can be used to compare the original virus with the green-lantern variant.
Response: We have now added a new subsection 3.1 at the beginning of the Results section to describe the construction of the new reporter virus and renumbered the following sections. Figure 1 has been revised to show the parental IndieC1 virus.
-Comment 7 : Figure 1 should be improved anyway, as the boxes representing the different open-reading frames are not correctly sized and positioned (most serious error: Rev 2 overlapping with green lantern instead of Tat2). All boxes should have a size reflecting the size of the orf, and they should be correctly positioned.
Response: Done.
-Comment 8 : Line 232: comparable? replication of the green-lantern virus is clearly slightly delayed compared to the original virus (3-days delay). This is no surprise and the authors could discuss probable causes (for example in the discussion).
Response: We did note that there was a small delay in the peak production. Therefore, we avoided the use of terms such as ‘similar’ or ‘identical’ but rather chose the term ‘comparable’ which is appropriate as the delay was small and the amount of peak virus production was also very similar. However, we have now revised the statement on line 232 from ‘comparable’ to indicate delayed replication. We have also added the following to the discussion section.
-Comment 9 : We observed a delay in virus replication for HIV-1IndieC1mGL when compared to the parental HIV-1IndieC1 virus (day 18 peak p24 production vs. day 15) in multi-day macrophage replication experiments (see Figure 3A and Figure 4A left panel). This is not surprising considering the two macro changes made to the virus genome, which may have affected some aspect of the genome RNA or any of the downstream functions of the genome with a consequent effect on the virus progeny population that manifested in a 3-day delay.
-Comment 10 : Line 237: what is meant with “the p24 to infectivity ratios”? Do the authors mean “the infectivity experiments?
Response: The quantity of p24 or the capsid protein in the medium determined via ELISA is a measure of the physical virus particle with no information about its activity. Infectivity determined as the amount of luciferase activity that results from using the virus in the medium to infect a reporter cell such as TZMbl is a measure of function. The p24 to infectivity ratio does not refer to infectivity experiments, but rather to how functional is the p24 we produced. We have now added an explanation - ‘relative luminescence units of luciferase activity to ng p24’ – in parentheses at the occurrence of the phrase “p24 to infectivity ratios”. Line : 259.
-Comment 11 : Line 262: via = measured by?
Response: The word ‘via’ is now replaced by ‘measured by’. : Line: 287.
-Comment 12 : Line 263: the use of the term MOI/multiplicity of infection is not correct. It is the amount of virus based on ng CA-p24 that was varied as input.
Rresponse: We understand the ambiguity felt by this reviewer. We agree that the concept of MOI in the classical sense was used as the number of virus particles per cell infected. However, this term in HIV research is widely used to indicate the ratio of virus to cells in multiple ways. We should have defined our definition of MOI. In the revised manuscript, we have included a sentence to correct this as follows: In this study, we define MOI as ng of p24 of HIV per 2x105 target cells. Line: 168.
-Comment 13 : Line 265-266: remove “was comparable”. Results/interpretations should be mentioned in the text, not in the legend. Moreover, the 2 viruses in B do NOT have a comparable infectivity.
Response: We have replaced the sentences as follows: “A. Infectivity of recombinant HIV-1IndieC1-mGL reporter virus measured alongside the parental HIV-1IndieC1. B. Infectivity of recombinant HIV-1NLAD8-GFP-Nef measured alongside HIV-1ADA.” Line: 290-291.
-Comment 14 : Line 283-284: 3 times, 3 separate donors, 2 replicates…… how does this result in n=3?
Response: We appreciate the reviewer’s emphasis on the use of accurate wording in our expressions. We have now included the following statement in the Methods section: “All the MDM experiments were performed three times (Figure 3, Figure 4 and Figure 5). Each experiment used MDMs from a separate donor and two experimental replicates were used to calculate an average. Thus, the data in each case are represented as mean ± SEM (n = 3).”Line: 183-184.
-Comment 15 : Line 302: as shown in our results >>> as shown in Figure… or remove
Response: The phrase ‘as shown in our results’ has been removed.
-Comment 16 : Line 309: we monitored replication upon infection of MDM….
Response: We agree. Corrected as follows: we also quantified mGreenLantern or GFP fluorescence respectively for cells infected with HIV-1IndieC1-mGL and HIV-1NLAD8-GFP-Nef viruses respectively. line : 362.
-Comment 17 : Line 319: ….. for each treatment for infection with…… this sentence is not clear
Response: We have now reworded this sentence as follows: “A fluorescence intensity plot for each treatment (e.g., +CCL2, +anti-CCL2 or +Isotype antibody control) for infection with each reporter virus (HIV-1IndieC1-mGL or HIV-1NLAD8-GFP-Nef)……” Line : 371.
-Comment 18 : Line 326: …. and by following the fluorescence pattern? Do the authors mean by quantification of the fluorescence?
Yes – this is what we meant. Now the word ‘following the fluorescence’ is replaced with ‘quantitating the mGreenLantern fluorescence” : Line: 379.
-Comment 19: Figures 4 and 5 are too small and can only be seen when enlarged on screen and not in a print.
Response : Both Figures have now been enlarged.
-Comment 20: Line 343. Data are represented….. Should this sentence be moved to the end of the legend? 3 donors, 2 replicates, how do the authors get to n=3?
Response : As mentioned above, this has already been explained above and a more detailed statement has been now added to methods.
-Comment 21: Line 355: in the presence of absence of Meth (Figure….). As indicated by the increased (?) levels of P24….
Response : We have now corrected the sentence as follows including the citation of the Figure 5: “As indicated by the increased levels of p24 measured at various times after infection up to day 15, Meth treatment enhanced the replication of all viruses compared to untreated HIV-infected MDMs (Figure 5).” Line: 389.
-Comment 21 : Line 363: 3.65 and 2.58 suggest very exact values…. 3.7 and 2.6 are more in line with 4.5. Same situation in line 367.
Response: Fixed. Line: 397.
-Comment 22: Line 384: remove differential
Response : We believe that the term differential explains that we are observing a differential effect of Meth in the two subtypes of HIV-1. This term has been retained.
-Comment 23: Line 385 and other positions in legend: subtype C is shown in the bottom panels and not in the top panels!
Response: This has been corrected.
-Comment 24 : Line 390 and 395: P: *, ,0.05? Also here explain n=3
Response: The extra comma is deleted. The explanation about n = 3 is now in the Methods section, as mentioned above.
-Comment 25: The authors should mention/discuss that, in their new virus, Nef is (or can be) produced from all spliced and unspliced viral transcripts (translation initiating at the IRES), and not only from the nef RNAs, which could result in an aberrant Nef production (see for example analysis of IRES-Nef constructs in Van der Velden et al. http://dx.doi.org/10.1016/j.virol.2015.11.004).
Response: We have added the following statement in the Discussion section: “The design of this reporter virus will facilitate translation of Nef protein from the primary, unspliced mRNA as well as from all spliced mRNAs due to the IRES inserted downstream of mGreenLantern.” Line: 439.
-Comment 26 : The authors could in the discussion also mention/discuss the stability of the introduced sequences in the virus (green lantern and IRES sequences will probably be lost upon long term culturing of the virus, but no problem in many short-term assays, like the assays shown in the paper).
Response: We have included the following section in the Discussion section: “While it is hard to pinpoint the mechanistic basis for this 3-day delay, one could imagine that the inserted sequences could affect the stability of chimeric viral genomic RNA. However, in the one month-long macrophage replication experiments we conducted (Figures 3 and 4), there were no observable defects.” Line: 448.
-Comment 27: Line 458: the authors state that the reporter virus has all the replication characteristics of its parental clone. That is true for the characteristics that they tested in the specific assays, but based on these assays one cannot say for all replication characteristics (as the introduced sequences will likely have all kind of not-identified effects, like an effect on Nef production, viral RNA splicing and possibly other processes).
Response : Modified as follows: “Although we did not perform an exhaustive series of tests, minimally, the reporter virus has retained the replication characteristics of its parental clone that were tested, which indicates that it can be used for cell culture and possibly in small animal models such as humanized mice.”
Reviewer 2 Report
Comments and Suggestions for Authors
In this paper the authors report the construction of a new HIV-1 subtype C virus with a green lantern reporter. This new virus present a macrophages tropism and can be used to study in vitro the replication of clade C virus and the impact on methamphetamine on their replication. The HIV field always needs new tools to study key questions in vitro and having a wild array of different subtype of virus is important. This new virus can be essential to numerous researchers however I think the characterization of this new construct and its potential to be used in different studies can be improved. This is why I would like to suggest a few comments below to hopefully built on and improve the work presented.
1- The title of the manuscript is extremely descriptive and narrows down the use of this new virus to replication in macrophages and study of the impact of meth. In my opinion by simply added a few experiments to your manuscript you would be able to call this manuscript the construction and characterization of a new subtype C HIV-1-m Green Lantern reporter virus. And let people choose what they will use this virus for without putting so much restriction.
2- I had issue finding a continuum in the ideas in this long introduction. I would have appreciate to read more about why we need a new HIV-C green and what are the limitations of the existing ones that the authors are trying overcome.
3- The method for the quantification of fluorescence intensity is missing important information like microscope used - settings to capture fluorescence intensity (microscope settings, average across well? sum? max?)- setting in imageJ - number or nuclei or area covered - any cell viability marker? control plasmid/transfected cells with both EGFP and Green lantern?
4- In the assessment of replication of the new virus the authors are using Lucifer's assay in TZM-bl cells and p24 concentration (MDM cells) and comparing the same virus without the GL reporter and an HIV clade-B with or without EGFP.
-With different scales on the graphs it is difficult to fully appreciate the difference of magnitude in replication between clade C and B.
-As expected, clade C present lower replication than B, however something really intriguing is completely ignored by the authors. the virus with the GL reporter present a clear delay in reaching peak production (18 vs 15) when compared to same virus without reporter (and even the clade B). This delay and slower ramp should be notify by the authors and investigated. It would be important to have the vRNA and cell associated RNA and DNA in those culture to see if this is due to a slower replication of the new construct or lower production of viral proteins and or artifact of calculating only p24 concentration.
- It would have been interesting to see data in CD4T cells too.
5-In the data set looking at the responsiveness to CCL2, first once again we see a delay to reach peak VL is delayed in the author's new virus which is not brought up in the manuscript. This experiment is missing information on cell viability especially since the authors are looking at p24 in supernatant and not vRNA.
To simplify the reading of the figures it would be great to indicate the virus used on the panel in the series of fluorescent pictures.
And while looking at fluorescence over time it would be good to have a control of the persistence of the GL and EGFP alone (transfected cells) over time.
6- In the methamphetamine effect assessment, the important data in those figures are the delta between treated and not treated so basically the statistic which are really small and hard to read- Would suggest the authors to present their data in fold change from no Meth, think it would make a better statement especially when compared to HIVNLAD8 and even more when looking at fluorescence intensity as the GL is 6 times brighter than EGFP so when comparing the bar graphs only we do not see any difference between the virus.
7- Ideally, since the authors are measuring the intensity of fluorescence it would have been more accurate to compare both viruses with both reporters (EGFP and GL) to really see the advantages and limitations of each constructs.
8- I would have been interested to see a longitudinal analysis of the reporters intensity to evaluate of GL is more stable over time than the EGFP which could help design in vitro studies with longer time. It would have also shown the advantages of using GL instead of EGFP. The authors aren't really stating why having a virus with GL would be better than using the existing EGFP ones.
Thanks for dedicating your time to developing new tools for the field.
Author Response
-Comment 1 : The title of the manuscript is extremely descriptive and narrows down the use of this new virus to replication in macrophages and study of the impact of meth. In my opinion by simply added a few experiments to your manuscript you would be able to call this manuscript the construction and characterization of a new subtype C HIV-1-m Green Lantern reporter virus. And let people choose what they will use this virus for without putting so much restriction.
Response : We would beg to differ from the reviewer. First, the title of the manuscript is not a diktat or a decree to our readers, but rather a descriptive title of what has been reported in the paper. We employed a macrophage-tropic virus (HIV-1IndieC1), which carries a envelope gene that is specialized for macrophage infection and replication. If a user wishes to study it in CD4 T cells, they would have to find a different reporter virus containing the appropriate envelope with the right tropism characteristics.
-Comment 2 : I had issue finding a continuum in the ideas in this long introduction. I would have appreciate to read more about why we need a new HIV-C green and what are the limitations of the existing ones that the authors are trying overcome.
Response : This is a good point. We have now emphasized that there is currently a single HIV-1 subtype C reporter virus in spite of the fact that subtype C is responsible for 50% of global HIV-1 burden.
-Comment 3 : The method for the quantification of fluorescence intensity is missing important information like microscope used - settings to capture fluorescence intensity (microscope settings, average across well? sum? max?)- setting in imageJ - number or nuclei or area covered - any cell viability marker? control plasmid/transfected cells with both EGFP and Green lantern?
Response : The type of microscope was already mentioned in the Methods section. Regarding control EGFP and mGreenlantern plasmids to be transfected into macrophages, these cells are hard to transfect.
-Comment 4 : In the assessment of replication of the new virus the authors are using Lucifer's assay in TZM-bl cells and p24 concentration (MDM cells) and comparing the same virus without the GL reporter and an HIV clade-B with or without EGFP.
Response: The initial luciferase assay was aimed at understanding whether the four virus preparations all displayed good infectivity to p24 (virus quantity) ratio. The subsequent p24 measurements in MDM cells are a means of finding out how much virus was released to the medium.
-Comment 5 : With different scales on the graphs it is difficult to fully appreciate the difference of magnitude in replication between clade C and B.
Response: It is true that subtype C HIV-1 replicates with reduced fitness compared to subtype B. Attempting to employ an identical scale would make the subtype C graph to be very small. Therefore, we opted to use different scales.
-Comment 6 : As expected, clade C present lower replication than B, however something really intriguing is completely ignored by the authors. the virus with the GL reporter present a clear delay in reaching peak production (18 vs 15) when compared to same virus without reporter (and even the clade B). This delay and slower ramp should be notify by the authors and investigated. It would be important to have the vRNA and cell associated RNA and DNA in those culture to see if this is due to a slower replication of the new construct or lower production of viral proteins and or artifact of calculating only p24 concentration.
Response : We agree with the comment that our GreenLantern construct replicates with a 3-day delay. This point was also raised by the Reviewer 1. We have now addressed this in the manuscript. Measuring p24 production is a standard procedure in every HIV-1 research laboratory and there are no known artifacts of measuring virus production. Moreover, we measure virus production from both the parental virus and the mGreenLantern virus.
-Comment 7 : It would have been interesting to see data in CD4T cells too.
Response : Macrophage-tropic viruses tend to replicate poorly on CD4 T cells and thus it would be of little value.
-Comment 8 : In the data set looking at the responsiveness to CCL2, first once again we see a delay to reach peak VL is delayed in the author's new virus which is not brought up in the manuscript. This experiment is missing information on cell viability especially since the authors are looking at p24 in supernatant and not vRNA.
Response: The delay has now been mentioned in the Results section as follows: “Like before, we observed a 3-day delay in the replication of HIV-1IndieC1mGL virus (virus production peak on 18-day instead of 15-day post-infection). We also included a subtype B control HIV-1NLAD8-GFP-Nef, which showed virus production at 15-day post-infection. Regarding the viability changes as a result of p24 production – no viability changes are expected. The p24 or Capsid protein is released as part of virus particles enclosed in viral membrane from the intact, infected cells by a budding process (similar to exocytosis) that leaves the cells intact. The cells are not lysed as a result of virus production.
-Comment 9 : To simplify the reading of the figures it would be great to indicate the virus used on the panel in the series of fluorescent pictures.
Response: Done
-Comment 10 : And while looking at fluorescence over time it would be good to have a control of the persistence of the GL and EGFP alone (transfected cells) over time.
Response: These controls would have required us to build separate non-HIV-1 vectors expressing these reporters separately. We did consider transfecting expression constructs, but transfection of macrophages is notoriously inefficient. However, we were satisfied that the fluorescence markers were active throughout the one month period of experiment in the case of macrophages and their expression correlated the active production of virus. Therefore, we did not pursue these experiments.
-Comment 11 : In the methamphetamine effect assessment, the important data in those figures are the delta between treated and not treated so basically the statistic which are really small and hard to read- Would suggest the authors to present their data in fold change from no Meth, think it would make a better statement especially when compared to HIVNLAD8 and even more when looking at fluorescence intensity as the GL is 6 times brighter than EGFP so when comparing the bar graphs only we do not see any difference between the virus.
Response: We would like point out that indeed, as the reviewer asked, we already did express the differences as fold-differences. Since the other reviewer also pointed out the readability of the figures, we have enlarged the sizes of Figures 4 and 5. Regarding the comment that the GreenLantern signal is 6-times brighter, we have added the following text to our revised manuscript: “First, we note that both viruses seem to display a similar level of fluorescence intensity despite the fact that mGreenLantern is 6-fold brighter than GFP. The reasons for this are that HIV-1NLAD8-GFP-Nef virus replicates much more robustly than HIV-1IndieC1mGL both because subtype B viruses replicate better than subtype C, but also because the chimeric nature of HIV-1NLAD8-GFP-Nef virus replicates to higher levels as mentioned above. We have therefore, made comparisons only within each virus at different levels of Meth used rather than between the viruses.”
-Comment 12: Ideally, since the authors are measuring the intensity of fluorescence it would have been more accurate to compare both viruses with both reporters (EGFP and GL) to really see the advantages and limitations of each constructs.
Response : The reviewer is recommending that we constructed multiple other reporters which was of a much bigger scope than the project completed here. We agree that we could have learnt a lot more by preparing these additional constructs.
-Comment 13 : I would have been interested to see a longitudinal analysis of the reporters intensity to evaluate of GL is more stable over time than the EGFP which could help design in vitro studies with longer time. It would have also shown the advantages of using GL instead of EGFP. The authors aren't really stating why having a virus with GL would be better than using the existing EGFP ones.
Response: We wanted to use the best, brightest GFP reporter virus. Therefore, we used GreenLantern. The 30 days culture of HIV-infected macrophages is the longest for HIV replication studies in most HIV investigations (with the exception of isolation of drug-resistant mutants or genetic revertants). Since our goal was to make available a subtype C HIV-1 reporter virus that would be suitable for myeloid-based studies, we have limited ourselves to this defined goal.
Round 2
Reviewer 2 Report
Comments and Suggestions for Authors
Thanks for addressing most of the comments brought up by reviewers.